# Extreme Learning Machine/Finite Impulse Response Filter and Vision Data-Assisted Inertial Navigation System-Based Human Motion Capture

**DOI:** 10.3390/mi14112088

**Published:** 2023-11-12

**Authors:** Yuan Xu, Rui Gao, Ahong Yang, Kun Liang, Zhongwei Shi, Mingxu Sun, Tao Shen

**Affiliations:** 1School of Electrical Engineering, University of Jinan, Jinan 250022, China; cse_xuy@ujn.edu.cn (Y.X.); 202221200878@stu.ujn.edu.cn (R.G.); 202130308035@stu.ujn.edu.cn (K.L.); 202130310075@stu.ujn.edu.cn (Z.S.); cse_sunmx@ujn.edu.cn (M.S.); 2School of Music, University of Jinan, Jinan 250022, China

**Keywords:** INS, vision, ELM, FIR, human position

## Abstract

To obtain accurate position information, herein, a one-assistant method involving the fusion of extreme learning machine (ELM)/finite impulse response (FIR) filters and vision data is proposed for inertial navigation system (INS)-based human motion capture. In the proposed method, when vision is available, the vision-based human position is considered as input to an FIR filter that accurately outputs the human position. Meanwhile, another FIR filter outputs the human position using INS data. ELM is used to build mapping between the output of the FIR filter and the corresponding error. When vision data are unavailable, FIR is used to provide the human posture and ELM is used to provide its estimation error built in the abovementioned stage. In the right-arm elbow, the proposed method can improve the cumulative distribution functions (CDFs) of the position errors by about 12.71%, which shows the effectiveness of the proposed method.

## 1. Introduction

In recent years, human motion capture has garnered considerable attention, owing to its diverse applications in entertainment, healthcare, and sports industries [1,2]. Accurate motion capture is essential for realistic animation, immersive virtual reality experiences, and a precise biomechanical analysis of human movements [3,4]. Traditional optical motion capture systems are widely employed, but they often exhibit certain limitations, such as high cost, restricted mobility, and dependency on controlled environments [5,6].

Many approaches have been proposed for human motion capture [7]. One of the most famous examples is the vision-based method. For instance, in [8], a vision-based system for tracking and interpreting leg motion in image sequences using a single camera is developed, which is implemented on a commercial computer without any special hardware. A new method for fast human motion capture based on a single red–green–blue-distance (RGB-D) sensor is proposed [9]. To the visual-based human posture capture device, Microsoft’s Kinect camera has gained popularity as a depth-sensing device that can capture human movements with high accuracy [10]. This camera utilizes infrared sensors to measure the distance between objects and the camera, generating a detailed three-dimensional point cloud representation of a scene [11]. This depth information, combined with RGB data, enables the precise tracking of human skeletal joints and facilitates real-time motion capture [12]. Depth cameras, integral to Kinect’s operation, rely on the emission and detection of infrared light to create depth maps of the surrounding scene [13,14]. However, the presence of environmental obstructions can introduce uncertainties into the captured data. Occurrences where the human body is temporarily occluded by objects within the view of the camera can lead to data gaps or inaccuracies in the motion capture process. These transient interruptions can impede the seamless reconstruction of motion trajectories, potentially affecting the fidelity and reliability of the captured human movement. In order to deal with this problem, inertial sensors have been proposed to measure human body movements. For instance, in [15], data on human activities are derived from the mobile device’s inertial sensor. Meanwhile, Beshara and Chen proposed the use of inertial sensors and Kinect cameras to capture human body movements in their study [16]. It should be noted that although the inertial sensor is able to achieve the seamless measurement, the sensors will experience cumulative errors.

Based on the measurement technology, the data fusion filter will also improve the accuracy of the measurement. To the data fusion filter, it should be pointed out that the Kalman filter (KF) has been widely used. For instance, in [17], the distributed Kalman filter has been proposed to provide a human’s position. The dual predictive quaternion Kalman filter is designed for the tracking of the human lower limb posture [18]. Moreover, one new adaptive extended Kalman filter (EKF) for cooperative localization is proposed [19], which is based on the nonlinear system. Moreover, the sigma-point update of cubature Kalman filter is proposed in [20]. One can easily find that the Kalman filter’s performance depends on the model’s accuracy and the noise description; however, it may be difficult to obtain in practice. In order to overcome this shortcoming, the finite impulse response (FIR) filter is proposed [21]. For example, the extended FIR (EFIR) is used to fuse the inertial navigation system (INS) data and the ultra-wide band (UWB) data. It should be pointed out that the approaches mentioned above do not consider the data outage, which may make the filter’s measurement unavailable. In order to overcome this problem, one least-squares support vector machine (LS-SVM)-assisted FIR filter is proposed. In [22], one self-learning square-root cubature Kalman filter is proposed for the integrated global positioning system (GPS)/INS in GPS-denied environments.

To address the limitations of standalone INS and overcome the data gaps in Kinect measurements [23,24], a previous study proposed the use of the extreme learning machine (ELM) algorithm to establish new signals through mapping when UWB signals are interrupted [25]. This allows the entire system to properly function. Building upon this concept, this paper proposes an integrated human motion capture system using ELM, FIR filtering, and INS data assisted by Microsoft’s Kinect camera [26], which can reconcile the strengths of INS and Kinect while reducing their weaknesses. The proposed methodology is outlined below.

The INS comprises miniature inertial sensors strategically placed on a subject’s body to measure accelerations and angular velocities [27,28]. Raw INS data provide real-time information about the orientation and motion of the subject [29]. The INS comprises miniature inertial sensors accurately placed on the subject’s body to measure attitude angles [30]. The raw INS data provide real-time information about the subject’s orientation and motion [31] and serve as a foundation for subsequent processes [32]. Meanwhile, the pivotal role of ELM lies in learning the intricate relation between INS-derived body pose data and the corresponding pose data acquired from Kinect [33,34]. Using a shallow neural network architecture, ELM efficiently maps the INS measurements to the corresponding Kinect-based body poses.

Before utilizing ELM, FIR is applied to both the INS data and the pose data obtained from Kinect [17,35]. This filtering process effectively eliminates sensor noise and mitigates the effects of drift, ensuring the accuracy and reliability of the motion capture system [36]. Finally, other researchers previously mentioned that the use of an interactive multiple model (IMM) filtering algorithm can further enhance positioning accuracy. Building upon this idea, the IMM filtering algorithm is employed. The IMM filter is adopted to fuse the INS data and the vision data from Kinect, alongside the ELM-processed data [37]. This fusion process compensates for the missing or erroneous Kinect measurements and further enhances the accuracy of the motion capture system [38,39].

By integrating INS, Kinect vision data, ELM algorithms, IMM algorithms, and FIR filtering, the proposed approach offers an advanced solution for human motion capture. This integration effectively reduces issues related to data gaps caused by environmental obstructions and high noise levels during Kinect measurements, contributing to improved precision and positioning accuracy in motion capture [40,41]. The resulting system ensures accurate and reliable real-time motion capture, thereby opening up a wide range of possibilities for applications in animation, virtual reality, sports analysis, and healthcare.

To obtain accurate position information, a one-assistant method fusing ELM/FIR filters and vision data is proposed for INS-based human motion capture. In the proposed method, when vision is available, the vision-based human position is inputted into an FIR filter that outputs accurate human position. Meanwhile, another FIR filter outputs the human position using INS data. Moreover, ELM is used to build mapping between the output of FIR and the corresponding error. When vision data are unavailable, FIR is used to provide human posture and ELM is used to provide the estimation error built in the aforementioned stage. Test results confirm the effectiveness of the proposed method.

The main contributions of this study are as follows:A seamless INS/vision human motion capture scheme is designed.A dual ELM/FIR-integrated filtering is derived.An INS/vision human motion capture system is built.Experimental evidence shows the better performances of the proposed algorithms than those of traditional algorithms.

The rest of this paper is structured as follows. Section 2 discusses the principle of an INS-based human motion capture system. Section 3 presents the design of an ELM/FIR filter for the human motion capture system. The investigation of experimental tests is summarized in Section 4, and the conclusions are given in Section 5.

## 2. INS-Based Human Motion Capture System

In this section of the study, the scheme of INS-based human motion capture is designed. The principle of the INS-based human motion capture system is illustrated in Figure 1. In the system used herein, we employ two types of sensors: IMU and vision. First, IMUs are fixed on a target person and employed to measure the posture of the target person’s torso. In this work, we employ eight IMUs to measure the acceleration and gyroscope values of the eight key joint points of the target human body. With the measured torso posture and corresponding torso length, the position of the target human’s joint points $L_{t}^{I,j},j\in \left[1,8\right]$ can be calculated. Meanwhile, we employ the Kinect-based vision sensor to measure the corresponding eight key joint points’ vision-derived position $L_{t}^{V,j},j\in \left[1,8\right]$.

From Figure 1, we can see that when vision data are available, the vision-derived position of the target human’s joint points $L_{t}^{V,j},j\in \left[1,8\right]$ can be obtained. Both $L_{t}^{I,j}$ and $L_{t}^{V,j}$ are input to the ELM/FIR filter, which is derived in the following section. Then, the ELM/EFIR filter outputs the estimated target human’s joints’ position $L_{t}^{j},j\in \left[1,8\right]$, which is used to compute the human posture. Notably, visual data are not continuously provided to the human pose measurement system in practical applications. When vision data are unavailable, only the $L_{t}^{I,j}$ are used by the ELM/FIR filter. Note that we only consider the human posture in this work, and the Kinect data may be difficult to obtain, which may result in data outage. It should be emphasized that a Kinect data outage of the eight joints does not occur simultaneously, but randomly. The main motivation of our proposed algorithm in this work is that it can provide relatively accurate pose information when the two types of sensors are working properly. Once visual information experiences an outage, it can also ensure the normal operation of the pose system.

### Calculation of the Human Joints’ Position

In this section, the calculation of the human joints’ position is proposed. First, the coordinate system and key parameters of the human body used herein are introduced. Second, the method of data measurement using the inertial and visual sensors is designed. Figure 2 shows the body frame (b-frame) and the navigation frame (n-frame) as well as the key parameters of the human body used herein. We employ the upper chest’s position as the coordinate origin. Only the positions of the elbows, wrist, knee, and ankle (${\mathrm{Po}}_{2}^{L},{\mathrm{Po}}_{2}^{R},{\mathrm{Po}}_{3}^{L},{\mathrm{Po}}_{3}^{R},{\mathrm{Po}}_{4}^{L},{\mathrm{Po}}_{4}^{R},{\mathrm{Po}}_{5}^{L},$ and ${\mathrm{Po}}_{5}^{R}$ shown in Figure 2) are considered. Here, we obtain ${\mathrm{Po}}_{1}^{L}=\left(x_{l}^{1},y_{l}^{1},z_{l}^{1}\right)$ and ${\mathrm{Po}}_{1}^{R}=\left(x_{R}^{1},y_{R}^{1},z_{R}^{1}\right)$. Thus, we can obtain the following:(1)$$\begin{array}{c} C_{n}^{b_{i}}=\left[\begin{array}{ccc} \cos\gamma_{i} & 0 & -\sin\gamma_{i}\\ 0 & 1 & 0\\ \sin\gamma_{i} & 0 & \cos\gamma_{i} \end{array}\right]\left[\begin{array}{ccc} 1 & 0 & 0\\ 0 & \cos\theta_{i} & \sin\theta_{i}\\ 0 & -\sin\theta_{i} & \cos\theta_{i} \end{array}\right]\left[\begin{array}{ccc} \cos\varphi_{i} & -\sin\varphi_{i} & 0\\ \sin\varphi_{i} & \cos\varphi_{i} & 0\\ 0 & 0 & 1 \end{array}\right]\\ =\left[\begin{array}{ccc} \cos\gamma_{i}\cos\varphi_{i}+\sin\gamma_{i}\sin\varphi_{i}\sin\theta_{i} & -\cos\gamma_{i}\sin\varphi_{i}+\sin\gamma_{i}\cos\varphi_{i}\sin\theta & -\sin\gamma_{i}\cos\theta_{i}\\ \sin\varphi_{i}\cos\theta_{i} & \cos\varphi_{i}\cos\theta_{i} & \sin\theta_{i}\\ \sin\gamma_{i}\cos\varphi_{i}-\cos\gamma_{i}\sin\varphi_{i}\sin\theta_{i} & -\sin\gamma_{i}\sin\varphi_{i}-\cos\gamma_{i}\cos\varphi_{i}\sin\theta_{i} & \cos\gamma_{i}\cos\theta_{i} \end{array}\right] \end{array}\;,$$
where $C_{n}^{b_{j}},j\in \left[0,8\right]$ indicates the rotation matrix from the b-frame of the *j*th point to the n-frame. θ, γ, and φ are pitch, roll, and yaw, respectively. Then, we can compute the joint’s position of the left and right upper arm (denoted as ${\mathrm{Po}}_{2}^{L}$ and ${\mathrm{Po}}_{2}^{R}$) using the following equations:(2)$$C_{b_{1}}^{b_{0}}=C_{n}^{b_{0}}C_{b_{1}}^{n}=C_{n}^{b_{0}}{\left(C_{n}^{b_{1}}\right)}^{T}\;,$$
(3)$${\mathrm{Po}}_{2}^{L}=l_{E}^{L}C_{b_{1},2}^{b_{0}}+{\mathrm{Po}}_{1}^{L}\;,$$
(4)$$C_{b_{1}^{R}}^{b_{0}}=C_{n}^{b_{0}}C_{b_{2}^{R}}^{n}=C_{n}^{b_{0}}{\left(C_{n}^{b_{2}^{R}}\right)}^{T}\;,$$
(5)$${\mathrm{Po}}_{2}^{R}=l_{E}^{R}C_{b_{1}^{R},2}^{b_{0}}+{\mathrm{Po}}_{1}^{R}\;.$$

Similarly, the position of the left and right wrist joints (denoted as ${\mathrm{Po}}_{3}^{L}$ and ${\mathrm{Po}}_{3}^{R}$) can be computed using the following equations:
(6)$$C_{b_{3}^{L}}^{b_{0}}=C_{n}^{b_{0}}C_{b_{3}^{L}}^{n}=C_{n}^{b_{0}}{\left(C_{n}^{b_{3}^{L}}\right)}^{T}\;,$$
(7)$${\mathrm{Po}}_{3}^{L}=l_{W}^{L}C_{b_{3}^{L},2}^{b_{0}}+{\mathrm{Po}}_{2}^{L}\;,$$
(8)$$C_{b_{3}^{R}}^{b_{0}}=C_{n}^{b_{0}}C_{b_{3}^{R}}^{n}=C_{n}^{b_{0}}{\left(C_{n}^{b_{3}^{R}}\right)}^{T}\;,$$
(9)$${\mathrm{Po}}_{3}^{R}=l_{W}^{R}C_{b_{3}^{R},2}^{b_{0}}+{\mathrm{Po}}_{2}^{R}\;.$$

We can compute the joint’s position of the left and right knee (denoted as ${\mathrm{Po}}_{4}^{L}$ and ${\mathrm{Po}}_{4}^{R}$) using the following equations:
(10)$$C_{b_{4}^{L}}^{b_{0}}=C_{n}^{b_{0}}C_{b_{4}^{L}}^{n}=C_{n}^{b_{0}}{\left(C_{n}^{b_{4}^{L}}\right)}^{T}\;,$$
(11)$${\mathrm{Po}}_{4}^{L}=l_{k}^{L}C_{b_{4}^{L},2}^{b_{0}}+{\mathrm{Po}}_{6}^{L}\;,$$
(12)$$C_{b_{4}^{R}}^{b_{0}}=C_{n}^{b_{0}}C_{b_{4}^{R}}^{n}=C_{n}^{b_{0}}{\left(C_{n}^{b_{4}^{R}}\right)}^{T}\;,$$
(13)$${\mathrm{Po}}_{4}^{R}=l_{k}^{R}C_{b_{4}^{R},2}^{b_{0}}+{\mathrm{Po}}_{6}^{R}\;.$$

Then, we can compute the joint’s position of the left and right ankle (denoted as ${\mathrm{Po}}_{5}^{L}$ and ${\mathrm{Po}}_{5}^{R}$) using the following equations:
(14)$$C_{b_{5}^{L}}^{b_{0}}=C_{n}^{b_{0}}C_{b_{5}^{L}}^{n}=C_{n}^{b_{0}}{\left(C_{n}^{b_{5}^{L}}\right)}^{T}\;,$$
(15)$${\mathrm{Po}}_{5}^{L}=l_{a}^{L}C_{b_{5}^{L},2}^{b_{0}}+{\mathrm{Po}}_{4}^{L}\;,$$
(16)$$C_{b_{5}^{R}}^{b_{0}}=C_{n}^{b_{0}}C_{b_{5}^{R}}^{n}=C_{n}^{b_{0}}{\left(C_{n}^{b_{5}^{R}}\right)}^{T}\;,$$
(17)$${\mathrm{Po}}_{5}^{R}=l_{a}^{R}C_{b_{5}^{R},2}^{b_{0}}+{\mathrm{Po}}_{4}^{R}\;.$$
where $C_{b_{j}^{L},2}^{b_{0}},C_{b_{j}^{R},2}^{b_{0}},j\in \left[0,8\right]$ represents the first column of $C_{b_{j}^{L}}^{b_{0}},C_{b_{j}^{R}}^{b_{0}},j\in \left[0,8\right]$.

## 3. ELM/FIR Integrated Filtering

In this section, we design ELM/FIR filtering, as shown in Figure 1 and Figure 2. First, we introduce the scheme of the ELM/FIR filter. Second, a data fusion model is derived. Third, the ELM and FIR methods designed based on the data fusion model are presented.

### 3.1. Scheme of the ELM/FIR-Integrated Filtering

In this work, we divide the combined filtering algorithm into two stages—training and prediction stages—when the Kinect data are available and unavailable, respectively.

The scheme of the ELM/FIR integrated filtering when Kinect data are available is shown in Figure 3. In this stage, we employ three FIR filters and two ELM methods. First, both the IMU and Kinect measure the joint position in parallel. Then, FIR filters 1 and 2 are used to estimate the positions of the IMU and Kinect. Thereafter, their estimations are fused using the IMM and Rauch–Tung–Striebel (R–T–S) smoothing method. Then, the output of the RTS smoothing method is used to compute the positions. Meanwhile, IMU’s and Kinect’s solutions are, respectively, used as the input and target of ELM 1, which is used to build the mapping between the IMU-measured position and the Kinect-measured position. The output of ELM 1 is used as the measurement of FIR filtering 3. ELM 2 employs the outputs of ELM 1 as input and the difference between the RTS method’s output and FIR filtering 3’s output as the target, which is used to build the mapping between them.

The scheme of the ELM/FIR-integrated filtering when Kinect data are unavailable is shown in Figure 4. In this stage, because we cannot obtain Kinect data, the IMU-measured position is input to ELM 1 directly. Then, ELM 1 outputs its estimated position with the mapping built in the training stage, which is employed by FIR filtering 3 as its measurement. Then, ELM 2 works to estimate the corresponding error, which is used to correct FIR filtering 3’s solution.

With all the joint points’ estimations, the positions can be computed. Notably, the Kinect data of each node may not necessarily be lost simultaneously.

### 3.2. Data Fusion Model of FIR Filtering

From the scheme mentioned above, we can see that three FIR filters are used in this work. The state equation used in this work can be given as follows:(18)$$\underset{L_{t}^{m,j-}}{\underset{︸}{\left[\begin{array}{c} {\mathrm{Lx}}_{t}^{m,j-}\\ {\mathrm{vx}}_{t}^{m,j-}\\ Ly_{t}^{m,j-}\\ vy_{t}^{m,j-} \end{array}\right]}}=\underset{T_{}^{m,j}}{\underset{︸}{\left[\begin{array}{cccc} 1 & \delta t & 0 & 0\\ 0 & 1 & 0 & 0\\ 0 & 0 & 1 & \delta t\\ 0 & 0 & 0 & 1 \end{array}\right]}}\underset{L_{t}^{m,j}}{\underset{︸}{\left[\begin{array}{c} {\mathrm{Lx}}_{t}^{m,j}\\ {\mathrm{vx}}_{t}^{m,j}\\ Ly_{t}^{m,j}\\ vy_{t}^{m,j} \end{array}\right]}}+w_{t}^{m,j}\;,$$
where $\left({\mathrm{Lx}}_{t}^{m,j-},{Ly}_{t}^{m,j-}\right)$ is the *j*th joint’s position at the time index *t*, $\left({\mathrm{vx}}_{t}^{m,j-},{vy}_{t}^{m,j-}\right)$ is the *j*th joint’s velocity at the time index *t*, *m* denotes the *m*th FIR filter used for the *j*th joint, δt is the sampling time, and $w_{t}^{m,j}∼N\left(0,Q\right)$ is the system noise.
(19)$$\underset{Z_{t}^{m,j}}{\underset{︸}{\left[\begin{array}{c} {L\hat{x}}_{t}^{m,j}\\ {L\hat{y}}_{t}^{m,j} \end{array}\right]}}=\underset{G}{\underset{︸}{\left[\begin{array}{cccc} 1 & 0 & 0 & 0\\ 0 & 0 & 1 & 0 \end{array}\right]}}L_{t}^{m,j-}+v_{t}^{m,j}\;,$$
where $\left({L\hat{x}}_{t}^{m,j},{L\hat{y}}_{t}^{m,j}\right),m\in \left[1,3\right]$ is the *j*th joint’s position measured using IMU, Kinect, and ELM 1’s output and $v_{t}^{m,j}∼N\left(0,R\right)$ is the measurement noise.

### 3.3. IMM-FIR Filtering

Based on models (Equation 18) and (Equation 19), the FIR filtering method used in this paper can be listed as Algorithm 1. In Algorithm 1, $D^{m,j}$ is the dimension of $L_{t}^{m,j}$, and $L^{m,j}$ is the filtering size. To the *m*th FIR filter, this algorithm has a dead zone. In this work, when the time index is in the dead zone, we employ the Kalman filer to replace the FIR’s work (lines 3–9 in Algorithm 1). When the time index is grater than the $L_{t}^{m,j}$, the FIR filer run the one-step prediction by using the following equations:(20)$$L_{jj}^{m,j-}=T^{m,j}L_{jj}^{m,j}+w_{jj}^{m,j}\;,$$

Then, the gain can be computed by using the following equation:
(21)$${H_{\mathrm{jj}}}^{m,j}={\left(GG^{T}+{\left(T^{m,j}H_{\mathrm{jj}-1}^{m,j}T^{m,j}\right)}^{-1}\right)}^{-1}\;,$$
(22)$$K_{jj}^{m,j}=H_{jj}^{m,j}G^{T}\;.$$

Then, the measurement update can be computed:
(23)$$L_{jj}^{m,j}=L_{jj}^{m,j-}+K_{jj}^{m,j}\left(Z_{jj}^{m,j}-GL_{jj}^{m,j-}\right)\;.$$

Note that the subscript jj is the counting of internal loops in FIR filter. The FIR filter employs the recent measurement to estimate the robust output.    
**Algorithm 1:** *m*th FIR filter used for the *j*th joint’s position in this work
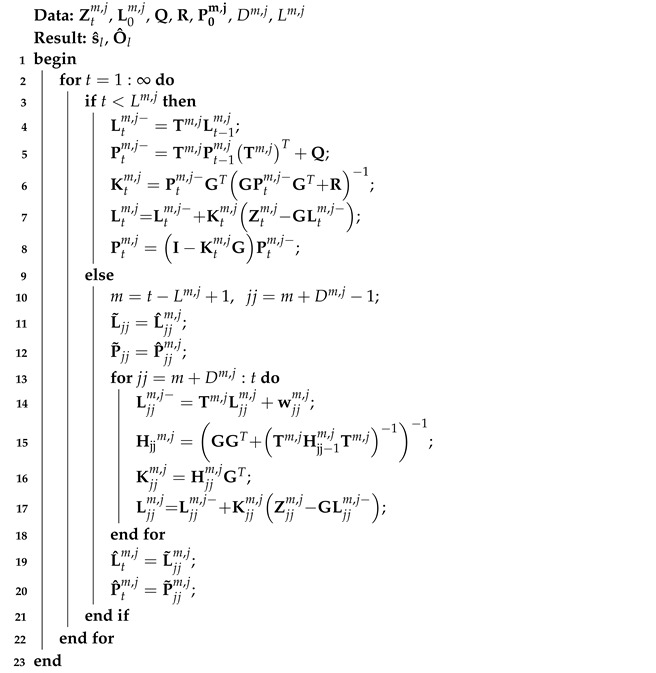


Figure 3 shows that we employ FIR filters 1 and 2 to measure the *j*th joint’s position $L_{t}^{1,j}$ and $L_{t}^{2,j}$ using IMU and Kinect’s data, respectively. Then, we introduce the *Markov* matrix, which can be given as follows:(24)$$\Gamma =\left[\begin{array}{cc} \Gamma_{11}^{} & \Gamma_{12}^{}\\ \Gamma_{21}^{} & \Gamma_{22}^{} \end{array}\right]\;.$$

Thus, we can determine that the predicted probability (normalization constant) ${\bar{c}}^{m}$ of model *i* to model *r* is:
(25)$${\bar{c}}^{m}=\sum_{i=1}^{r}\Gamma_{i}^{m}\mu_{i}^{m,j-},m\in \left[1,2\right]\;.$$

The mixed probability $\mu_{i}^{m,j-}$ from model *i* to model *r* is:
(26)$$\mu_{i}^{m,j-}=\sum_{i=1}^{r}\Gamma_{i}\mu_{i}^{m,j-}/{\bar{c}}^{m},m\in \left[1,2\right]\;,$$
where μ is the model probability for each filter and Γ is the probability transfer matrix.

With the FIR filters 1 and 2, we can determine that the blended state estimation of the model ${\hat{L}}_{0}^{m,j-}$ and the mixed covariance estimation of the model $P_{0}^{m,j-}$:
(27)$${\hat{L}}_{0}^{m,j-}=\sum_{i=1}^{r}{\hat{L}}_{i}^{m,j-}\mu_{i}^{m,j-}\;,$$
(28)$$P_{0}^{m,j-}=\sum_{i=1}^{r}\mu_{i}^{m,j-}\left(P_{i}^{m,j-}+\left({\hat{L}}_{i}^{m,j-}-{\hat{L}}_{0}^{m,j-}\right){\left({\hat{L}}_{i}^{m,j-}-{\hat{L}}_{0}^{m,j-}\right)}^{T}\right)\;.$$

Take ${\hat{L}}^{m,j}$, $P^{m,j-}$, and $Z_{t}$ as inputs to update prediction state ${\hat{L}}_{t}^{m,j}$ and filter covariance $P_{t}^{m,j}$:
(29)$${\hat{L}}^{1,j}=T^{1,j-}{\hat{L}}_{0}^{1,j-}\;,$$
(30)$${\hat{L}}^{2,j}=T^{2,j-}{\hat{L}}_{0}^{2,j-}\;,$$
(31)$$P^{1,j-}=T^{1,j-}P_{0}^{1,j-}{\left[T^{1,j-}\right]}^{T}+Q\;,$$
(32)$$P^{2,j-}=T^{2,j-}P_{0}^{2,j-}{\left[T^{2,j-}\right]}^{T}+Q\;,$$
(33)$$K_{t}^{1}=P^{1,j-}G^{T}{\left(GP^{1,j-}G^{T}+R\right)}^{-1}\;,$$
(34)$$K_{t}^{2}=P^{2,j-}G^{T}{\left(GP^{2,j-}G^{T}+R\right)}^{-1}\;,$$
(35)$${\hat{L}}_{t}^{1,j}={\hat{L}}^{1,j-}+K_{t}^{1}\left(Z_{t}^{1,j}-G{\hat{L}}^{1,j}\right)\;,$$
(36)$${\hat{L}}_{t}^{2,j}={\hat{L}}^{2,j-}+K_{t}^{2}\left(Z_{t}^{2,j}-G{\hat{L}}^{2,j}\right)\;,$$
(37)$$P_{t}^{1,j}=\left(I-K_{t}^{1}G\right)P^{1,j-}\;,$$
(38)$$P_{t}^{2,j}=\left(I-K_{t}^{2}G\right)P^{2,j-}\;.$$

The likelihood function $\Lambda_{t}^{i}$ is used to update the model probability $\mu_{t}$, calculated as follows:
(39)$$\Lambda_{t}^{1}=\frac{1}{{\left(2\pi \right)}^{n/2}{\left|S_{t}^{1}\right|}^{1/2}}\exp\left\{-\frac{1}{2}{\left(v_{t}^{1}\right)}^{T}{\left(S_{t}^{1}\right)}^{-1}v_{t}^{1}\right\}\;,$$
(40)$$\Lambda_{t}^{2}=\frac{1}{{\left(2\pi \right)}^{n/2}{\left|S_{t}^{2}\right|}^{1/2}}\exp\left\{-\frac{1}{2}{\left(v_{t}^{2}\right)}^{T}{\left(S_{t}^{2}\right)}^{-1}v_{t}^{2}\right\}\;,$$
where $v_{t}^{i}$ and $S_{t}^{i}$ are the measurement error and measurement error covariance matrix, calculated as follows:
(41)$$v_{t}^{1}=Z_{t}^{1,j}-T^{1,j}{\hat{L}}_{t,j}^{1}\;,$$
(42)$$S_{t}^{1}=GP^{1,j}G^{T}+R\;,$$
(43)$$v_{t}^{2}=Z_{t}^{2,j}-T^{1,j}{\hat{L}}_{t}^{2,j}\;,$$
(44)$$S_{t}^{2}=GP^{2,j}G^{T}+R\;,$$
where $\mu_{t}$ is the credibility after fusion and *c* is the denominator coefficient to achieve normalization, calculated as follows:
(45)$$c=\sum_{m=1}^{r}\Lambda_{t}^{m}{\bar{c}}^{m}$$
(46)$$\mu_{t}^{1}=\Lambda_{t}^{1}\bar{c}/c\;.$$

Based on the updated confidence level $\mu_{t}^{m}$, the models can be fused sequentially to obtain the output target state and covariance matrix of the algorithm:
(47)$${\hat{L}}_{t}^{m,j}=\sum_{m=1}^{r}{\hat{L}}^{m,j}\mu_{t}^{m}\;,$$
(48)$$P_{t}^{m,j}=\sum_{j=1}^{r}\mu_{t}^{m}\left(P_{j}^{m,j}+\left({\hat{L}}_{j}^{m,j}-{\hat{L}}^{m,j}\right){\left({\hat{L}}_{j}^{m,j}-{\hat{L}}^{m,j}\right)}^{T}\right)\;.$$

### 3.4. ELM Method

The ELM method used in this work is similar to the method we proposed previously [42]. In this section, we introduce this method briefly. On the basis of the data fusion model mentioned above, we can compute ELM’s activation function $\chi \left(\cdot \right)\;$ as follows:(49)$$\sum_{q=1}^{d}\alpha_{q}\chi \left(\beta_{q}I_{e}+b_{q}\right)=\delta_{e},e\in \left[1,s-1\right]\;,$$
where $I_{e}=\left[I_{1},I_{2},...,I_{s-1}\right]$. Thus, we can obtain the following equation:(50)$$\sum_{q=1}^{d}\alpha_{q}\chi \left(\beta_{q}I_{e}+b_{q}\right)=y_{e},e\in \left[1,s-1\right]\;.$$

Then, (50) can be rewritten as
(51)$$\underset{F_{}^{E}}{\underset{︸}{\left[\begin{array}{c} f_{1}^{E}\left(I_{1}\right)\\ \vdots \\ f_{s-1}^{E}\left(I_{s-1}\right) \end{array}\right]}}\underset{\beta^{E}}{\underset{︸}{\left[\begin{array}{c} \beta_{1}\\ \vdots \\ \beta_{s-1} \end{array}\right]}}=\underset{z^{E}}{\underset{︸}{\left[\begin{array}{c} z_{1}\\ \vdots \\ z_{s-1} \end{array}\right]}}\;,$$
and $\beta^{E}$ can be computed using the following equation:(52)$${\hat{\beta }}^{E}=F_{}^{E+}z^{E}\;.$$

## 4. Test

In this work, we perform one real test to show the effectiveness of the proposed method. First, the corresponding experimental parameters are presented. Then, the performance of the proposed method is investigated.

### 4.1. Experimental Parameters

Experimental parameters are introduced in this section. Figure 5 shows the block diagram of the experimental system. Herein, we employed eight IMUs fixed on the target human to measure the corresponding posture. Meanwhile, Kinect was also used to measure the target human’s joint positions. All the sensors’ data were collected using a computer. In this work, the model of the IMU is ICM-20948, and the IMU parameters are given in Table 1.Only eight joint positions were considered herein, as shown in Figure 1. Meanwhile, we employed Kinect 2.0 as the visual sensor in this work; its parameters are given in Table 2. The test environment is shown in Figure 6. When we performed the test, the target human moved in front of Kinect and the IMUs were fixed on the target human’s body. The human moved according to predetermined movements, and the computer collected all the sensor data.

### 4.2. Positioning Accuracy

In this section, six joints’ positioning accuracies measured using the proposed method were investigated. Herein, we employed the solutions of IMU, Kinect, and ELM/KF filtering to compare their performances. Figure 7 displays the positions measured using Kinect, IMU, ELM/FIR, and ELM/KF in the right-arm elbow. In this figure, the black line represents the solution of the ELM/FIR method, the blue line represents the solution of the ELM/KF method, the green line represents the solution of the IMU-only solution, the pink line represents the solution of the Kinect-only solution, and the red line represents the reference value. The figure shows that in this joint, the performances of ELM/FIR and ELM/KF were similar, and the positions estimated using the methods mentioned above were between the outputs of the Kinect and IMU. It can be seen that in the figure, the blue and black lines are almost identical in all three directions. In the figure, it can be seen that there are some protrusions in the solution values of the ELM/FIR and ELM/KF filters, which indicate that the Kinect data are not available in these time indexes. However, when Kinect data were unavailable, the output of the proposed ELM/FIR method was closer to the solution of the ELM/KF method, which deduced that the proposed ELM/FIR method was effective in maintaining the performance of the filter during Kinect data outage.

The cumulative distribution functions (CDFs) of the position errors measured using ELM/FIR and ELM/KF in the right-arm elbow are shown in Figure 8. In this figure, the red line represents the solution of the ELM/FIR method, the blue line represents the solution of the ELM/KF method. As shown in this figure, the proposed ELM/FIR method exhibited better performance than that of ELM/KF. At 0.9, the proposed ELM/FIR filter could reduce the localization error from 0.0866 to 0.0678 m, improving the localization accuracy by approximately 12.71%, which indicates that the proposed ELM/FIR is more effective than the proposed ELM/KF method in this joint.

The positions measured using Kinect, IMU, ELM/FIR, and ELM/KF in the right-arm wrist are listed in Figure 9. Similar to the case of the right-arm elbow, when the data of IMU and Kinect are available, the performance of the ELM/FIR and ELM/KF are almost the same. And when the Kinect data experience an outage, it can bee seen that the black lines are closer to the reference value compared to the blue line, which means that the proposed method is more effective in reducing the localization error when Kinect data are unavailable. Figure 10 displays the CDF of the position errors measured using ELM/FIR and ELM/KF in the right-arm wrist. The proposed method thus improved the localization error by approximately 16.40%. Moreover, the root-mean-squared errors (RMSEs) measured using the ELM/FIR and ELM/KF filters in the right-arm wrist are listed in Table 3. The table shows that the ELM/FIR method improved the localization error from 0.0690 to 0.0641 m compared to the ELM/KF method. From the figures and the table mentioned above, it can be seen that the proposed ELM/FIR is more effective than the proposed ELM/KF method in this joint.

The CDFs of the position errors measured using ELM/FIR and ELM/KF in the left-arm elbow are shown in Figure 11. Therein, the position error of ELM/FIR is smaller than that of ELM/KF. This shows that the proposed method is effective in reducing the positioning error.

The positions measured using Kinect, IMU, ELM/FIR, and ELM/KF in the left-arm wrist are given in Figure 12. Similar to the case of the right-arm elbow, the proposed method is more effective in reducing the localization error when Kinect data are unavailable. Figure 13 displays the CDF of the position errors measured using ELM/FIR and ELM/KF in the left-arm wrist. The proposed method thus improved the localization error by approximately 56.71%. Moreover, the RMSEs measured using the ELM/FIR filter and ELM/KF filter in the left-arm wrist are presented in Table 4. According to this table, the ELM/FIR method improved the localization error from 0.1493 to 0.0607 m compared with the ELM/KF method.

The positions measured using Kinect, IMU, ELM/FIR, and ELM/KF in the right knee are shown in Figure 14. Similar to the case of the right-arm elbow, the proposed method is more effective in reducing the localization error when Kinect data are unavailable. Figure 15 displays the CDF of the position errors measured using ELM/FIR and ELM/KF in the right knee. The proposed method improved the localization error by approximately 48.32%. Moreover, the RMSEs measured using the ELM/FIR and ELM/KF filters in the right knee are given in Table 5. The table shows that the ELM/FIR method improved the localization error from 0.1128 to 0.0566 m compared with the ELM/KF method.

The position error CDFs measured using ELM/FIR and ELM/KF in the right ankle are presented in Figure 16. This figure shows that the proposed method’s solution is closer to the reference value. Meanwhile, the CDF results show that the proposed method substantially improved the positioning accuracy.

The RMSEs measured using the ELM/FIR and ELM/KF filters in the left knee are shown in Figure 17. Herein, the RMSEs of the two filters in the *x* and *y* directions are similar. However, the proposed method shows its effectiveness in the *z* direction.

From the analysis of the positioning accuracies of the different joints mentioned above, we can conclude that the proposed method is effective in reducing localization error, especially in Kinect outage areas.

## 5. Conclusions

To obtain accurate position information, a one-assistant method fusing the ELM/FIR filter and vision data was proposed herein for INS-based human motion capture. In the proposed method, when vision is available, the vision-based human position inputs to an FIR filter that accurately outputs the human position. Meanwhile, another FIR filter outputs the human position using INS data. Moreover, ELM was used to build mapping between the output of FIR and the corresponding error. When vision data were unavailable, FIR was used to provide the human posture and ELM was used to provide the estimation error built in the previously mentioned stage. In order to show the effectiveness of the proposed ELM/FIR filter, eight joint points are considered in this work. Test results show that the localization errors of the eight joint points measured using the proposed ELM/FIR filter are smaller than the values of the ELM/KF filter, especially when a Kinect data outage occurs, which demonstrates the effectiveness of the proposed method.

## Figures and Tables

**Figure 1 micromachines-14-02088-f001:**
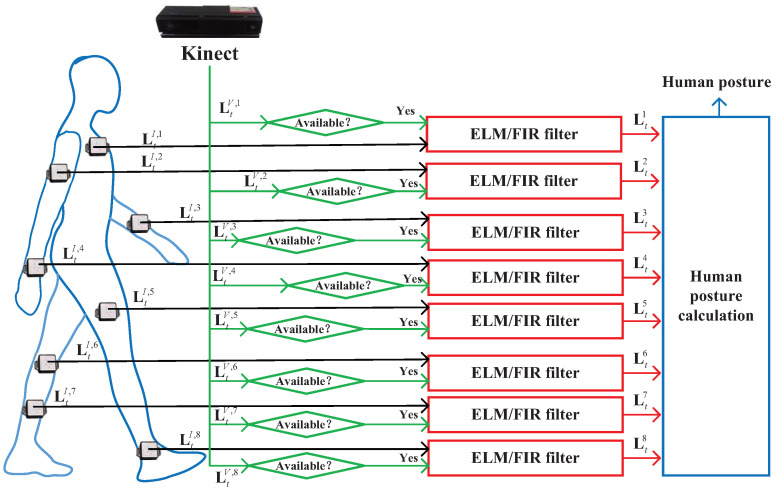
Principle of the INS-based human motion capture system.

**Figure 2 micromachines-14-02088-f002:**
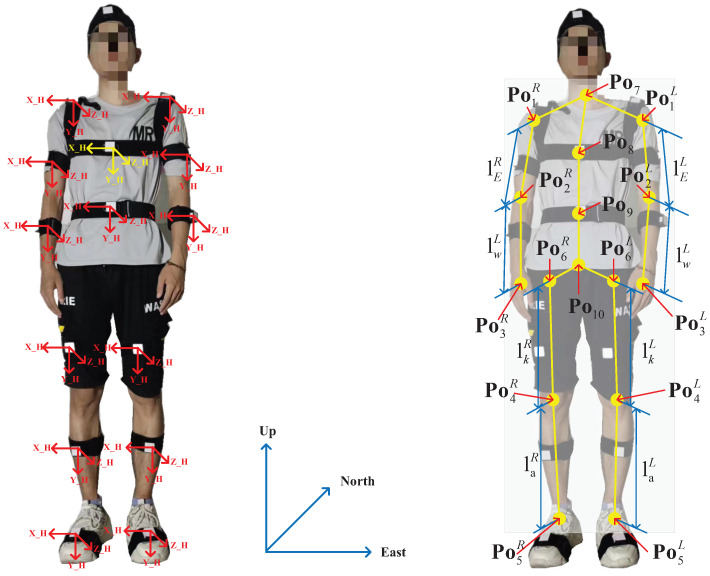
Coordinate system and key parameters of the human body used in this work.

**Figure 3 micromachines-14-02088-f003:**
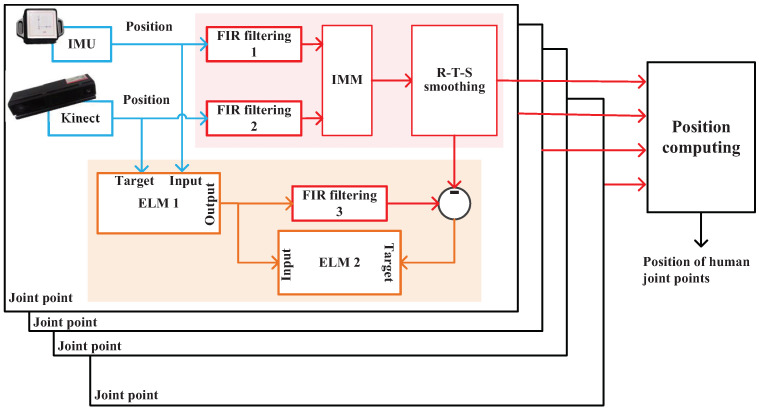
Scheme of the ELM/FIR-integrated filtering when Kinect data are available.

**Figure 4 micromachines-14-02088-f004:**
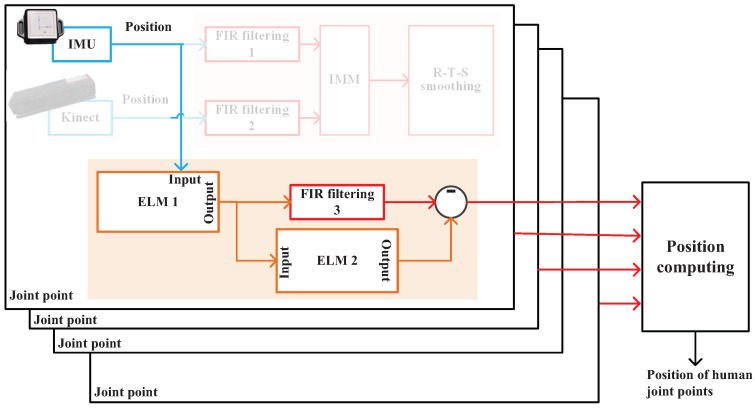
Scheme of the ELM/FIR-integrated filtering when Kinect data are unavailable.

**Figure 5 micromachines-14-02088-f005:**
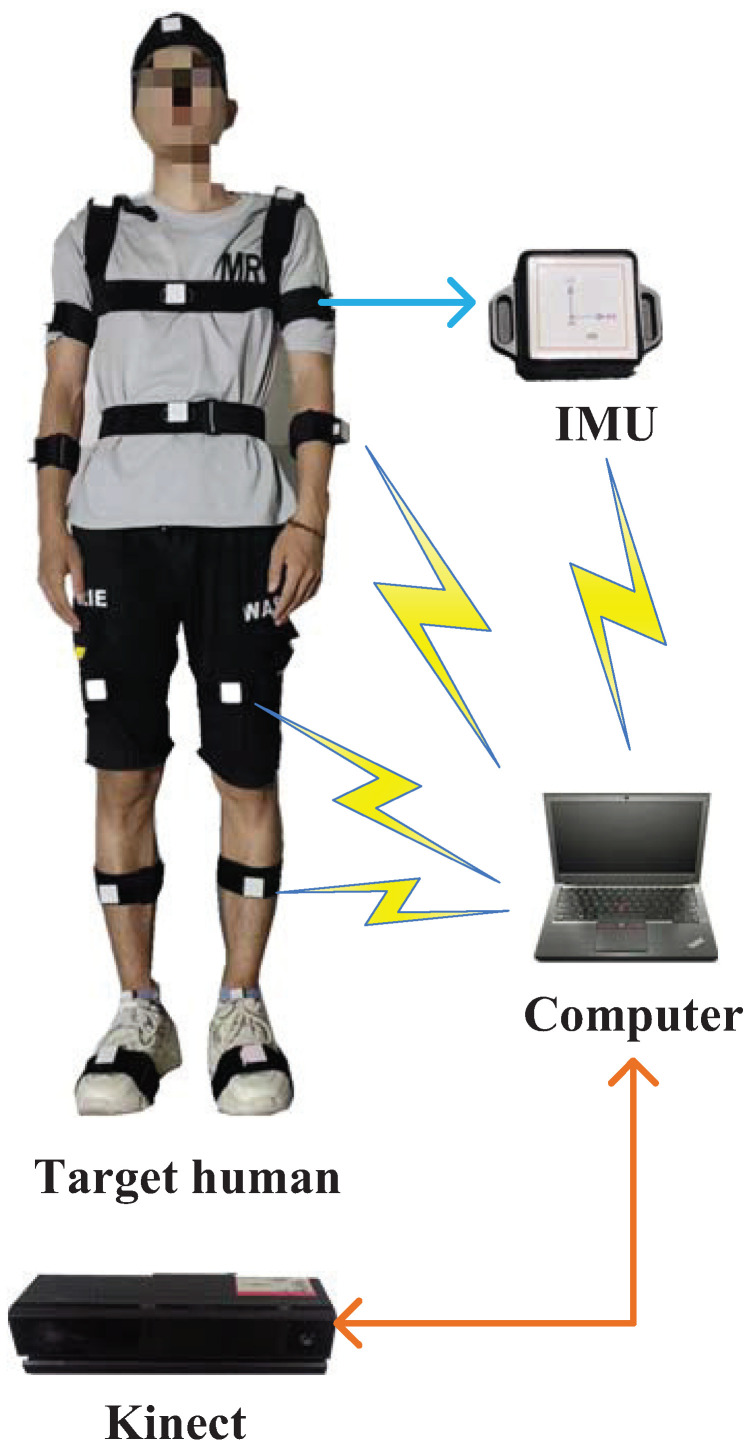
Block diagram of the experimental system.

**Figure 6 micromachines-14-02088-f006:**
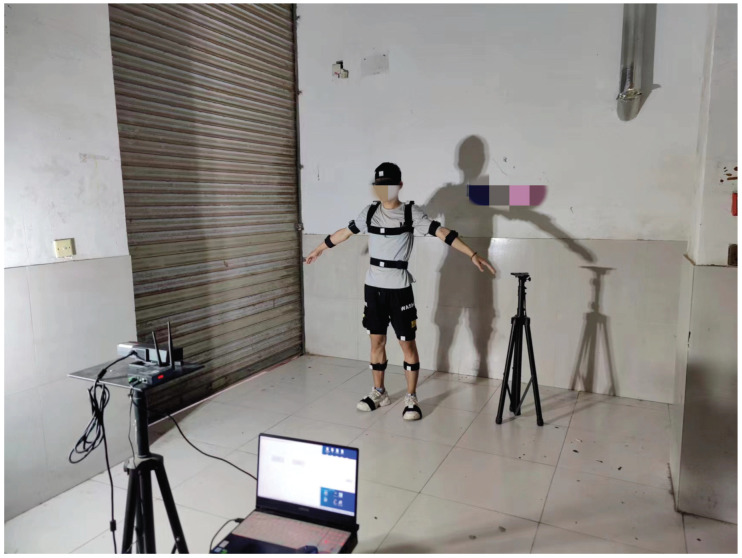
Test environment.

**Figure 7 micromachines-14-02088-f007:**
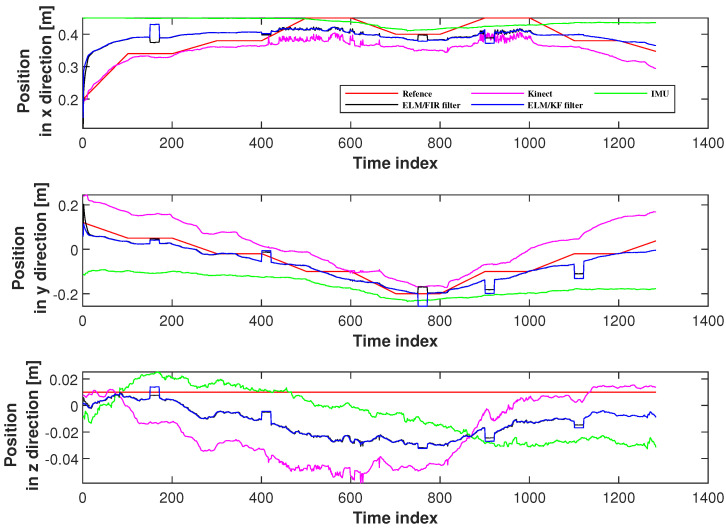
Positions measured using Kinect, IMU, ELM/FIR, and ELM/KF in the right-arm elbow.

**Figure 8 micromachines-14-02088-f008:**
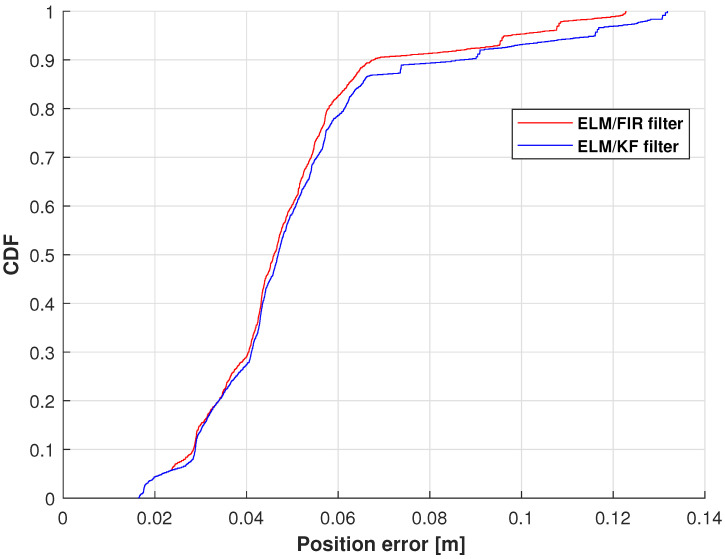
CDF of the position errors measured using ELM/FIR and ELM/KF in the right-arm elbow.

**Figure 9 micromachines-14-02088-f009:**
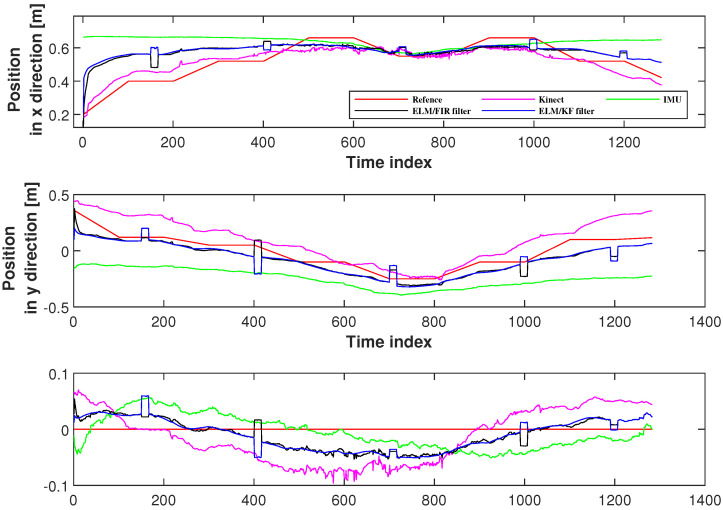
Positions measured using Kinect, IMU, ELM/FIR, and ELM/KF in the right-arm wrist.

**Figure 10 micromachines-14-02088-f010:**
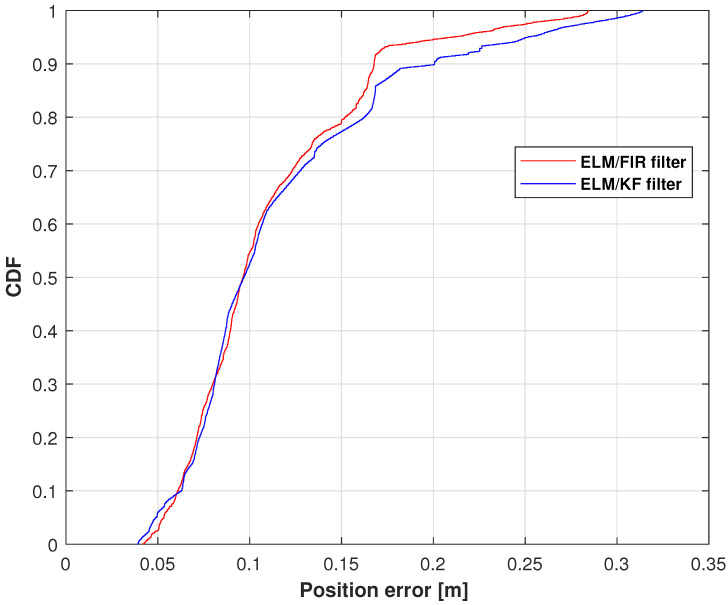
CDF of the position errors measured using ELM/FIR and ELM/KF in the right-arm wrist.

**Figure 11 micromachines-14-02088-f011:**
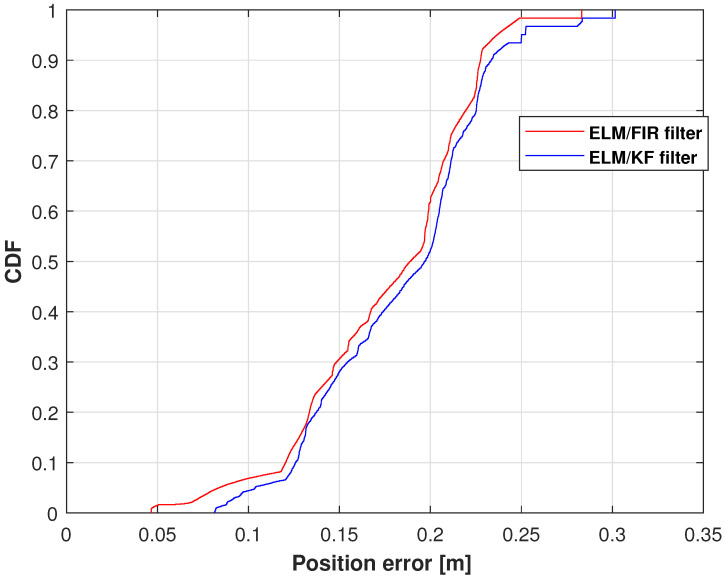
CDF measured using ELM + FIR and ELM + KF in the left-arm elbow.

**Figure 12 micromachines-14-02088-f012:**
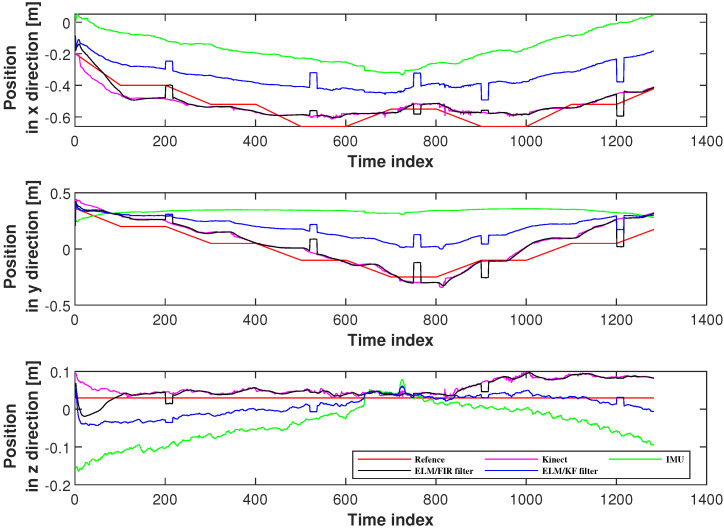
Positions measured using Kinect, IMU, ELM/FIR, and ELM/KF in the left-arm wrist.

**Figure 13 micromachines-14-02088-f013:**
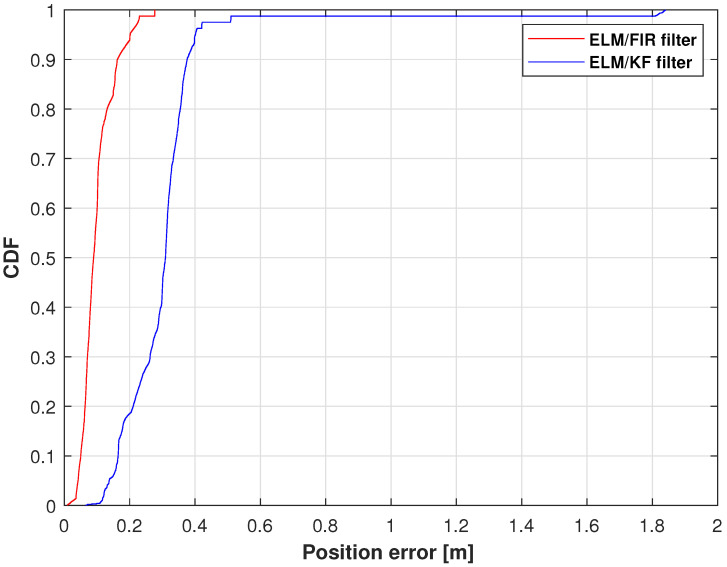
CDF measured using ELM + FIR and ELM + KF in the left-arm wrist.

**Figure 14 micromachines-14-02088-f014:**
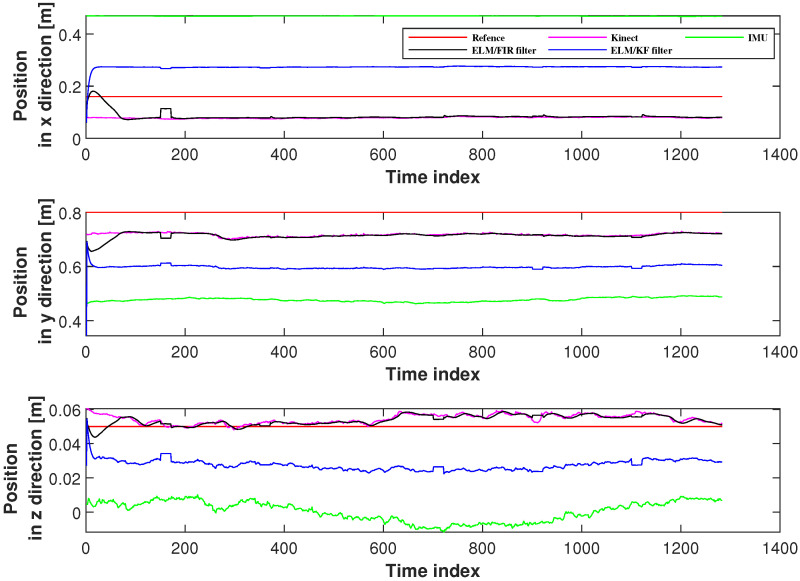
Positions measured using Kinect, IMU, ELM/FIR, and ELM/KF in the right knee.

**Figure 15 micromachines-14-02088-f015:**
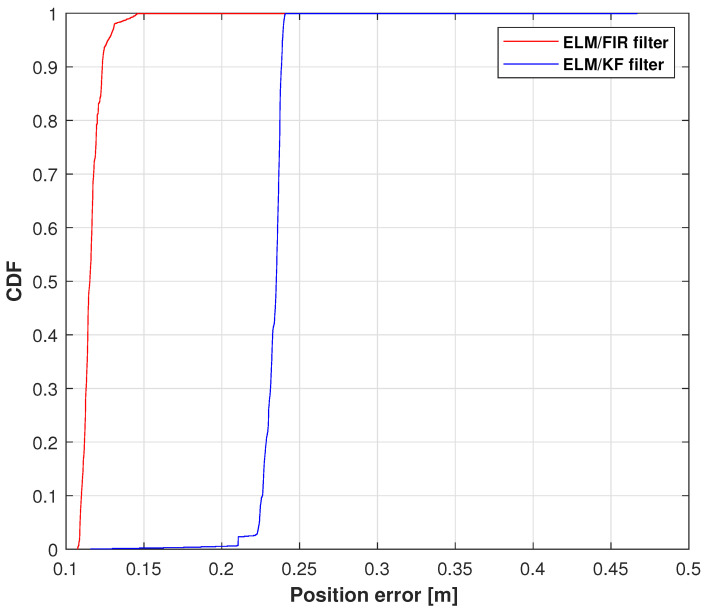
CDF measured using the ELM/FIR and ELM/KF filters in the right knee.

**Figure 16 micromachines-14-02088-f016:**
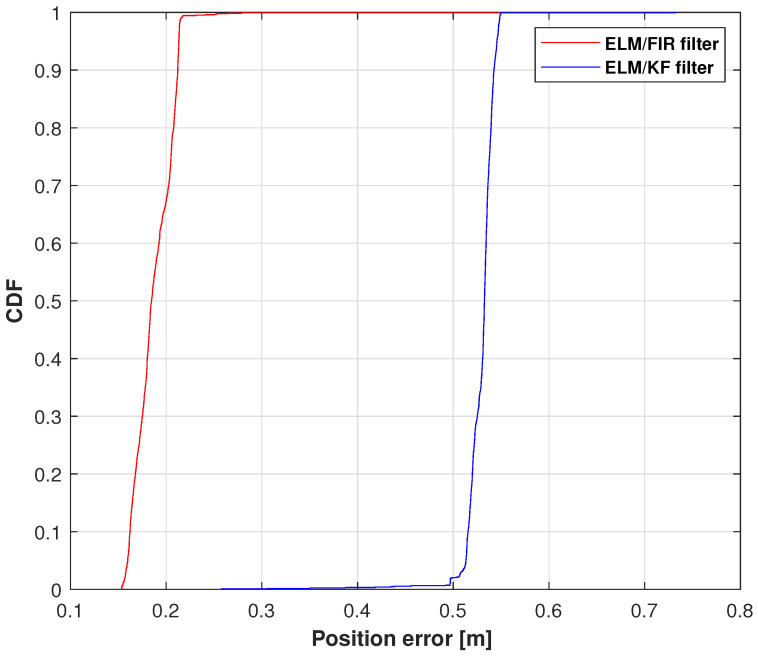
CDF measured using the ELM/FIR and ELM/KF filters in the right ankle.

**Figure 17 micromachines-14-02088-f017:**
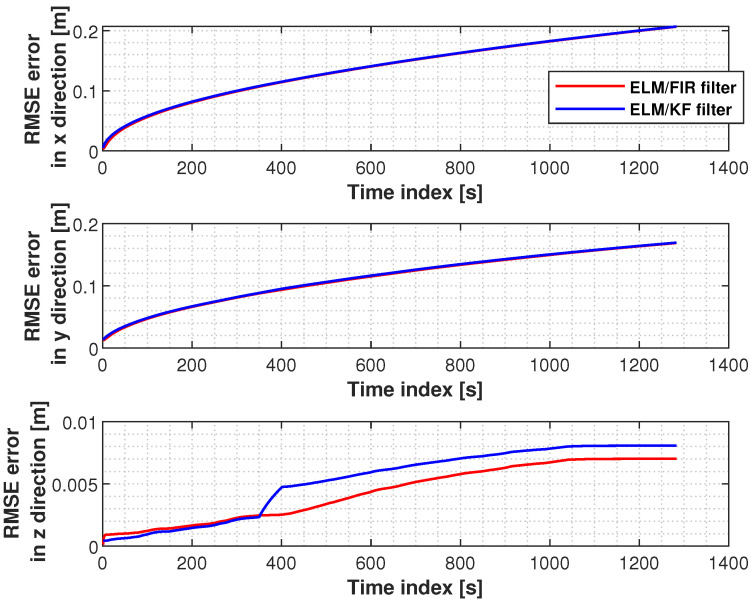
RMSEs measured using the ELM/FIR and ELM/KF filters in the left knee.

**Table 1 micromachines-14-02088-t001:** Parameters of the IMUs used in the test.

Parameter	Value
Sensor’s precision	$0.01^{\circ }$ (pitch and roll), $0.1^{\circ }$ (yaw)
Sampling frequency	100 Hz
Measurement dimension	3
Data transmission distance	100 m
Working voltage	4.2 V

**Table 2 micromachines-14-02088-t002:** Parameters of Kinect used in the test.

Parameter	Value
Resolution of color image frames	1920×1080
Resolution of deep frames	512×424
Detectable range	0.5–4.5 m
Resolution of infrared image frames	512×484
Field of view	$70^{\circ }\times 60^{\circ }$

**Table 3 micromachines-14-02088-t003:** RMSEs measured using the ELM/FIR and ELM/KF filters in the right-arm wrist.

Methods	X (m)	Y (m)	Z (m)	Mean (m)
ELM/KF filter	0.0959	0.0819	0.0292	0.0690
ELM/FIR filter	0.0903	0.0739	0.0282	0.0641

**Table 4 micromachines-14-02088-t004:** RMSEs measured using the ELM/FIR filter and ELM/KF filter in the left-arm wrist.

Methods	X (m)	Y (m)	Z (m)	Mean (m)
ELM/KF filter	0.2023	0.2103	0.0352	0.1493
ELM/FIR filter	0.0617	0.0873	0.0333	0.0607

**Table 5 micromachines-14-02088-t005:** RMSEs measured using the ELM/FIR and ELM/KF filters in the right knee.

Methods	X (m)	Y (m)	Z (m)	Mean (m)
ELM/KF filter	0.1133	0.2025	0.0226	0.1128
ELM/FIR filter	0.0771	0.0880	0.0047	0.0566

## Data Availability

Data are contained within the article.

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
