# Peer review of "Extreme Learning Machine/Finite Impulse Response Filter and Vision Data-Assisted Inertial Navigation System-Based Human Motion Capture"

_micromachines, 2023, doi:10.3390/mi14112088_

Round 1

Reviewer 1 Report

Comments and Suggestions for Authors

Some comments are listed for the improvement of this research at this stage:

(1) The introduction of the last paragraph of Section I is not coincident with the real part of the Sections 2, 3 and so on;

(2) The joints positon calculation from Equations (2)to (17) cannot express all the joints in Human body.

(3) Figures (1) and (2) should input together and explain their differences.

(4) The ELM/FIR integrated filtering scheme in Figure 4 seem some problems, such as  the ELM2 without the output ? why ?

(5) The IMUs used in the test with Stablization precision 0.01° ? What is this mean ?  Which type of IMU used ?

(6) Most of the Symbol used in the Equations are not explained clealy.

(7) ELM/FIR is compared with ELM/KF  in Section 4, why ?

Comments on the Quality of English Language

The overall English should be improved by native English speaker.

Author Response

Dear Reviewers,

Thank you very much for valuable comments, which helped the authors to improve the manuscript. Following the reviewers’ remarks and suggestions, the manuscript have been carefully revised to improve the quality for your further consideration.

Comment 1

The introduction of the last paragraph of Section I is not coincident with the real part of the Sections 2, 3 and so on;

Answer:

I am sorry for my mistake. We have corrected this mistake in the revised paper.

Comment 2

The joints positon calculation from Equations (2)to (17) cannot express all the joints in Human body.

Answer:

Thank you for your insightful comment. In revised paper, we have reorganized the wording of this section.

Comment 3

Figures (1) and (2) should input together and explain their differences.

Answer:

Thank you. We have combined Fig. 1 and Fig. 2, and in the revised work, only Fig. 1 has been retained, and a detailed explanation of the positioning strategy has been provided.

Comment 4

The ELM/FIR integrated filtering scheme in Figure 4 seem some problems, such as the ELM2 without the output ? why ?

Answer:

The Fig. 4 has been transformed into Fig.3 in the revised paper, in this stage, the ELM 2 is in training stage, in this stage, ELM 2 employs the outputs of ELM 1 as input and the difference between the R-T-S method’s output and FIR filtering 3’s outputs as the target, which is used to build the mapping between them. In this stage, the ELM 2 just work to build the mapping, do not output any value.

Comment 5

The IMUs used in the test with Stablization precision 0.01° ? What is this mean ? Which type of IMU used ?

Answer:

I am sorry for my mistake. I have already revised this mistake in the revised paper. In this work, the model of the IMU is ICM-20948, and the IMU parameters are given in Tab. 1. Here, the sensor’s precision means the accuracy of the pitch, roll, and yaw.

Comment 6

Most of the Symbol used in the Equations are not explained clealy.

Answer:

Thank you for your insightful comment. This problem has been revised.

Comment 7

ELM/FIR is compared with ELM/KF in Section 4, why ?

Answer:

In this work, the motivation for employing the ELM/KF as the benchmark is to show the effectiveness of the proposed ELM/FIR method, from the test results, we can see that the proposed ELM/FIR is effective to reduce the localization error, especially when Kinec data is outage.

Reviewer 2 Report

Comments and Suggestions for Authors

In this paper, the authors propose a one-assistant method involving fusion of extreme-learning machine (ELM)/finite-impulse response (FIR) filters and vision data is proposed for inertial navigation system (INS)-based human motion capture.  The paper is well written and well organized, and I have some other suggestions that can help the paper serve the reader better: 

1. Lack of introduction to the data fusion filtering and its development in the introduction.

2. The contribution of this paper should be refined, and I encourage the author to add the results of the method  (quantitative) in the abstract.

3. Please explain the motivation for using the combination of IMU and Kinect. 

4. Whether the running speed is affected after multiple types of integrated navigation are used?

5. If the test is online?

Comments on the Quality of English Language

See above.

Author Response

Dear Reviewers,

Thank you very much for valuable comments, which helped the authors to improve the manuscript. Following the reviewers’ remarks and suggestions, the manuscript have been carefully revised to improve the quality for your further consideration.

Comment 1

Lack of introduction to the data fusion filtering and its development in the introduction.

Answer:

Thank you. This problem has been revised.

Comment 2

The contribution of this paper should be refined, and I encourage the author to add the results of the method (quantitative) in the abstract.

Answer:

Good idea. We add the results of the method (quantitative) in the abstract.

Comment 3

Please explain the motivation for using the combination of IMU and Kinect.

Answer: To obtain accurate position information, herein, a one-assistant method involving fusion of ELM/FIR filters and vision data is proposed for INS-based human motion capture. In the proposed method, when vision is available, the vision-based human position is considered input to an FIR filter that accurately outputs the human position. Meanwhile, another FIR filter outputs the human position using INS data. ELM is used to build mapping between the output of the FIR filter and the corresponding error. When vision data are unavailable, FIR is used to provide the human posture and ELM is used to provide its estimation error built in the abovementioned stage.

Comment 4

Whether the running speed is affected after multiple types of integrated navigation are used?

Answer:

Thank you, it should be pointed out that the proposed method will affect the running speed, however, with the advancement of hardware devices, the impact of the proposed ELM/FIR algorithm on runtime can be overcome. At the same time, although the algorithm proposed in this article is multi-level, algorithms at different levels can run in parallel and the training scale is not large, which lays the foundation for reducing computational time.

Comment 5

If the test is online?

Answer:

I am sorry, the test used in this work is online, however, it should be pointed out that the way we process data is offline.

Round 2

Reviewer 1 Report

Comments and Suggestions for Authors

I recommended the acceptance of this manuscript with the updated version.

Comments on the Quality of English Language

English is fine, depth improvement would be much better.